# Synthesis and Structure of the Bis- and Tris-Polyhedral Hybrid Carboranoclathrochelates with Functionalizing Biorelevant Substituents—The Derivatives of Propargylamine Iron(II) Clathrochelates with Terminal Triple C≡C Bond(s)

**DOI:** 10.3390/molecules26123635

**Published:** 2021-06-14

**Authors:** Genrikh E. Zelinskii, Ilya P. Limarev, Anna V. Vologzhanina, Valentina A. Olshevskaya, Anton V. Makarenkov, Pavel V. Dorovatovskii, Alexander S. Chuprin, Mikhail A. Vershinin, Semyon V. Dudkin, Yan Z. Voloshin

**Affiliations:** 1Kurnakov Institute of General and Inorganic Chemistry of the Russian Academy of Sciences, Leninskii pr., 31, 119991 Moscow, Russia; heinrichzelinskiy@gmail.com (G.E.Z.); limarev.1995@mail.ru (I.P.L.); 2Nesmeyanov Institute of the Organoelement Compounds of the Russian Academy of Sciences, Vavilova Str., 28, 119991 Moscow, Russia; vologzhanina@mail.ru (A.V.V.); olshevsk@ineos.ac.ru (V.A.O.); anton_makarenkov@mail.ru (A.V.M.); alschuprin@gmail.com (A.S.C.); sdudkin@ineos.ac.ru (S.V.D.); 3National Research Center Kurchatov Institute, 1 Kurchatova pl., 123098 Moscow, Russia; paulgemini@mail.ru; 4Nikolaev Institute of Inorganic Chemistry of the Siberian Branch of the Russian Academy of Sciences, 3 Lavrentieva prosp., 630090 Novosibirsk, Russia; mvershinin@ngs.ru

**Keywords:** clathrochelates, carboranes, polyhedral compounds, iron complexes, ligand reactivity, “click” reactions, 1,3-dipolar cycloaddition

## Abstract

A synthetic strategy for obtaining structurally flexible hybrid iron(II) carboranoclatrochelates functionalized with biorelevant groups, based on a combination of a 1,3-dipolar cycloaddition reaction with nucleophilic substitution of an appropriate chloroclathrochelate precursor, was developed. In its first stage, a stepwise substitution of the dichloroclathrochelate precursor with amine *N*-nucleophiles of different natures in various solvents was performed. One of its two chlorine atoms with morpholine or diethylamine in dichloromethane gave reactive monohalogenoclathrochelate complexes functionalized with abiorelevant substituents. Further nucleophilic substitution of their remaining chlorine atoms with propargylamine in DMF led to morpholine- and diethylamine-functionalized monopropargylamine cage complexes, the molecules of which contain the single terminal C≡C bond. Their “click” 1,3-cycloaddition reactions in toluene with *ortho*-carborane-(1)-methylazide catalyzed by copper(II) acetate gave spacer-containing di- and tritopic iron(II) carboranoclatrochelates formed by a covalent linking between their different polyhedral(cage) fragments. The obtained complexes were characterized using elemental analysis, MALDI-TOF mass, UV-Vis, ^1^H, ^1^H{^11^B}, ^11^B, ^11^B{^1^H}, ^19^F{^1^H} and ^13^C{^1^H}-NMR spectra, and by a single crystal synchrotron X-ray diffraction experiment for the diethylamine-functionalized iron(II) carboranoclathrochelate. Its encapsulated iron(II) ion is situated almost in the center of the Fe*N*_6_-coordination polyhedron possessing a geometry intermediate between a trigonal prism and a trigonal antiprism with a distortion angle *φ* of approximately 28°. Conformation of this hybrid molecule is strongly affected by its intramolecular dihydrogen bonding: a flexibility of the carborane-terminated ribbed substituent allowed the formation of numerous C–H…H–B intramolecular interactions. The H(C) atom of this carborane core also forms the intermolecular C–H…F–B interaction with an adjacent carboranoclathrochelate molecule. The N–H…N intermolecular interaction between the diethylamine group of one hybrid molecule and the heterocyclic five-membered *1H*-[1,2,3]-triazolyl fragment of the second molecule of this type caused formation of H-bonded carboranoclathrochelate dimers in the X-rayed crystal.

## 1. Introduction

Among the earlier-described *d*-metal monomacrobicyclic complexes (clathrochelates [1,2]) and their bis-cage analogs, shown in Scheme 1 with terminal biorelevant, first of all, carboxyl group(s), compounds **1**–**5** are reported to be the most prospective bioeffectors, including the so-called “topological drugs” [2,3] and molecular optical probes [4,5,6]. They possess the highest inhibitory activities in the transcription systems of T7 RNA [7,8] and Taq DNA [9] polymerases, the best antifibrillogenic properties [10] and the most intensive CD outputs on their supramolecular binding to albumins [4,5,6] as well. The latter results are explained [4,5,6] by strong supramolecular interactions of the terminal polar and/or H-donor group(s) of a given macrobicyclic “guest” with appropriate aminoacid residues of a suitable protein macromolecule as a “host” upon their non-covalent host–guest clathrochelate–protein self-assembly. Moreover, very recently, one of the macrobicyclic iron(II) complexes, the molecule that contains one terminal functionalizing carboxyl group, has been found [11] to possess both high in vitro cytotoxicity and selectivity (as compared with normal cells) against the human promyelocytic leukemia cell line. Earlier, we observed [12] the similar effect of an electrophilic iron(II) hexachloroclathrochelate **6** that easily alkylated glutathione in these cells. It is clear that such a mechanism of cytotoxic activity of cage complexes with an encapsulated metal ion is substantially less probable in the case of metal clathrochelates, the molecules of which do not contain the highly reactive terminal group(s) or atom(s), first of all, the halogen atom(s) as the ribbed substituent(s) in a macrobicyclic framework. However, these functionalizing substituents in the chelate fragment(s) of highly π-conjugated polyazomethine encapsulating ligands are known [1,2] to substantially affect the spatial and electronic structure of their quasiaromatic cage complexes and, therefore, their chemical reactivity and redox properties. The most effective and convenient pathway of a ribbed functionalization of these complexes is based on nucleophilic substitution of their appropriate reactive halogenoclathrochelate precursors with various *N, O, S, C*-nucleophilic agents [2].

However, the polyhedral boron and boron–carbon molecular scaffolds, such as closo-borates [13], carboranes [14] and their functionalized derivatives, are available, chemically robust and low-toxicity compounds. Therefore, they are regarded [15,16,17,18,19,20,21,22,23] as the most promising precursors of the boron-enriched targets for ^10^B-neutron capture therapy (NCT) treatment for a wide range of cancers. The key problem with their practical medicinal application in such types of radiotherapy is a target delivery of the above boron-enriched compounds to the tumor cells of a given cancer, thus allowing a decrease in the amounts that are required for effective ^10^B-NCT treatment and preventing additional intoxication of the patient’s body. The use of the spacer-containing hybrid carboranoclatrochelates, formed by a covalent linking between the different polyhedral (cage) fragments of the same polytopic molecule, seems to be very prospective from this point of view. Indeed, their chemical design allows the targeted regulation of a hydrophilic–hydrophobic balance in the corresponding hybrid molecules using various clathrochelate (as a more polar core)-to-carborane (as a more hydrophobic core) stoichiometries, as well as the efficient supramolecular binding to a given biological target using the variation of their spatial structure and their functionalization with biorelevant substituent(s) at these macrobicyclic (polyhedral) subparts of a given hybrid molecule.

One of the feasible and convenient approaches to obtain the hybrid carboranoclathrochelates with flexible spacer fragment(s) between their polyhedral(cage) subparts is based on the use of a 1,3-dipolar [2+3] cycloaddition reaction [24,25] of an appropriate carborane-based azide to an alkyne-terminated clathrochelate precursor. The azide group is known [26,27] to be the most convenient 1,3-dipole, which almost does not produce unfavorable by-products: it possesses a chemical robustness against water and air oxygen, while the formed *1H*-[1,2,3]-triazole fragment does not undergo the metabolic transformations [28]. It should also be noted that such 1,3-cycloaddition proceeds under mild reaction conditions; moreover, it is not sensitive to the presence of other reactive functionalizing groups, such as hydroxyl or amine substituent(s), in the molecules of the corresponding 1,3-dipole and dipolarophile components. The use of an appropriate copper salt as its catalyst, as well as the rational choice of a solvent, allows this “click” reaction to be successfully performed not only under the above mild conditions, but also in relatively high or moderate yields. Earlier, the monomethinomonopropargylamine iron(II) cage complex **7** (Scheme 1) was successfully used [29] as a model alkyne-terminated macrobicyclic precursor for its ribbed functionalization with a flexible carborane-terminated substituent using a copper-catalyzed “click” reaction of this precursor to give carboranoclathrochelate **8**. The presence of only one terminal C≡C group in its molecule allowed [29] to avoid unfavorable side reactions and, therefore, the formation of hardly separable admixtures of the corresponding by-products. Thus, the developed synthetic approach, in its combination with a stepwise nucleophilic substitution of an appropriate dichloromacrobicyclic precursor, allowed us to obtain the reactive monohalogenoclathrochelate complexes containing one functionalizing biorelevant substituent for strong supramolecular binding to biomacromolecules, and then their macrobicyclic derivatives with the single terminal C≡C group for further transformations of these cage intermediates using a 1,3-dipolar cycloaddition reaction, giving a series of polytopic and multicage covalent conjugates—the structurally flexible hybrid iron(II) carboranoclatrochelates.

## 2. Results and Discussion

Our synthetic strategy for obtaining of the titled compounds, functionalized with a biorelevant ribbed substituent, used, for its first stage, a stepwise nucleophilic substitution of the dichloroclathrochelate precursor FeBd_2_(Cl_2_Gm)(BF)_2_ with amine *N*-nucleophiles of different natures in various solvents. Indeed, the performed [30] detailed study of nucleophilic substitution of the iron(II) polychloroclatrochelates with aliphatic amines as *N*-nucleophilic agents allowed evaluation of the effects of both the chemical nature of a given amine nucleophile and that of a solvent used (i.e., its donor ability, polarity and H-donor properties) on a chemical composition of the clathrochelate products of this reaction. In particular, an introduction of the first aliphatic NH-containing substituent into a halogen-containing chelate cycle of the corresponding clathrochelate precursor with ribbed dichloroclatrochelate fragment(s) instead of one of its two chlorine atoms (this means passing from this atom as an inherent ribbed substituent to the aliphatic secondary amine group) activates the second chlorine atom in a *vic*-position of the same α-dioximate fragment. As a result, in the case of the hexachloroclatrochelateiron(II)-encapsulating precursors, the next stage of their nucleophilic substitution with aliphatic primary amines proceeded through a formation of the corresponding diamine ribbed fragment of a macrobicyclic encapsulating ligand. Conversely, an introduction of the first aliphatic tertiary amine substituent instead of one chlorine atom as an inherent ribbed substituent deactivates the remaining (second) chlorine atom into a *vic*-position of the same chelate fragment of an encapsulating ligand, and it does not undergo nucleophilic substitution even in the presence of a high excess of the corresponding secondary aliphatic amine, and even in media containing highly donor and polar solvents, such as DMF and DMSO. Therefore, we chose diethylamine and morpholine, the secondary aliphatic amines, as the *N*-nucleophiles leading to the iron(II) monochloroclatrochelate precursors, the molecules of which are functionalized with one biorelevant ribbed substituent: the corresponding amine residues are known to form strong supramolecular bonds with amino acid chains of protein macromolecules.

Nucleophilic substitution of one of the two chlorine atoms of the iron(II) dichloroclatrochelate **9**, as a macrobicyclic precursor, with morpholine or diethylamine, was performed by Scheme 2 in dichloromethane as a solvent. Further nucleophilic substitution of the remaining chlorine atom of the obtained monochloroclathrochelates **10** and **11,** with propargylamine as a more active primary amine *N*-nucleophile in DMF as a highly donor and polar solvent, led to the morpholine- and diethylamine-functionalized monopropargylamine cage complexes **12** and **13**, the molecules of which contain the single terminal C≡C bond (Scheme 3).

The next stage of our synthetic strategy used the “click” reactions of the obtained alkyne-terminated amine-containing macrobicyclic precursors. Toluene was chosen as the most suitable solvent for this reaction because of its chemical robustness (i.e., its substantially lower chemical reactivity) and the absence of donor ability, as compared with those of acetonitrile, alcohols, water, and other polar and/or H-donor solvents. Therefore, toluene does not form strong solvato complexes with copper ions or with copper-containing intermediates. The obtained monopropargylaminemacrobicyclic complexes underwent in this solvent the 1,3-dipolar cycloaddition reactions shown in Scheme 4 with *ortho*-carborane-(1)-methylazide catalyzed by copper(II) acetate. Earlier, we found [18,31] that copper(I)-based catalysts are insufficiently active for efficient performing of such 1,3-cycloaddition reactions with terminal acetylenes. Attempts using *N*-donor compounds such as bipyridine or ethylenediamine derivatives as the co-ligands allowing activation of copper(I) salts, in the case of *ortho*-carborane-(1)-methylazide as a dipolarophile component, caused formation of various by-products, the molecules of which contain a *nido*-carborane core. However, a high catalytic activity of copper(I) and copper(II) salts with more acceptor counter-ions, such as acetate, triflate and trifluoroacetatemonoanions, has been observed [32]; the catalytic activity of a series of copper(I) and copper(II) compounds in this reaction between the regular organic azides and acetylenes has been compared [33]. The obtained results suggest that copper(II) acetate is most active among them; we also found [18,31] that this cooper(II) salt efficiently catalyzes the cycloaddition reactions of *ortho*-carborane-(1)-methylazide with terminal acetylenes. In order to avoid formation of the complexes {Cu_2_(μ-C≡CR)_2_}, which are reported [33,34] to give the acetylene dimerization products in the presence of air oxygen, the “click” reactions under study were performed in an inert atmosphere (under argon). We also performed the analogous double-“click” reaction by Scheme 5 of a known [35] dipropargylamine iron(II) clathrochelate **16** as a bifunctionalmacrobicyclic precursor with two terminal C≡C bonds.

The obtained complexes were characterized using elemental analysis, MALDI-TOF mass, UV-Vis, ^1^H, ^1^H{^11^B}, ^11^B, ^11^B{^1^H}, ^19^F{^1^H} and ^13^C{^1^H}-NMR spectra, and also by a single crystal X-ray diffraction experiment (for the hybrid diethylamine-functionalized carboranoclathrochelate **15**).

MALDI-TOF mass spectra of these compounds contain in their positive range the peaks of the corresponding molecular ions and those of their ionic associates with Na^+^ and K^+^ ions with characteristic isotopic distributions, which were in good agreement with those theoretically calculated.

^1^H, ^19^F and ^13^C{^1^H}-NMR spectra of the obtained monocage precursors and their hybrid polytopic derivatives confirmed the structure and symmetry of their molecules. The spectra of the monopropargylamine iron(II) clathrochelates and their monocarboranomonoclathrochelare derivatives contain the characteristic signals of a functionalizing propargylamine substituent with a terminal C≡C group and those with a carborane core with attached *1H*-[1,2,3]-triazolyl-containing spacer fragments, respectively. The integral intensities (in the ^1^H-NMR spectra) and the number of the signals (in the ^1^H, ^19^F and ^13^C{^1^H}-NMR spectra) confirmed the chemical drawings of these molecules. It should be noted that the ^1^H-NMR signal of protons of a spacer methylene group between a carborane polyhedron and the above triazole cycle in the spectra of the hydride iron(II) carboranoclathrochelates appeared as a singlet at approximately 5.0 ppm, thus suggesting a H-acidic character of this group.

Their ^11^B-NMR spectra appear as a superposition of the ^11^B-NMR spectra of the earlier-described [24,25] carborane-terminated triazoles, containing signals in the range δ^11^_B_ = −0.9 ÷ −14.5 ppm, and those of the fluoroboron-capped iron(II) clathrochelates, containing characteristic doublet(s) at approximately 3.60 ppm of their O_3_BF capping groups, caused by the spin–spin ^11^B—^19^F interactions in them.

The results of deconvolution of the UV–vis spectra of the obtained di- and monopropargylamineclathrochelates, their monoclathrochelate precursors and the hybrid carboranoclathrochelate derivatives into their Gaussian components are compiled in Appendix A (see SI). The UV–vis spectrum of the dichloroclathrochelate precursor **9** contains two intensive Fe*d* → Lπ * metal-to-ligand charge transfer (MLCT) bands in the visible range with maxima at approximately 450 and 470 nm. Nucleophilic substitution of one of its ribbed chlorine atoms, giving the functionalizing biorelevant amine substituent at a cage quasiaromatic framework, resulted in the monochloromonoamine cage complexes **10** and **11**, in the spectra of which a contradirectional shift of these MLCT bands and an appearance of a new longwave band in the range 500–550 nm are observed. Further functionalization of these iron(II) monochloroclathrochelate precursors using their nucleophilic substitution with propargylamine and using the “click” 1,3-dipolar cycloaddition reaction resulted in the appearance of one more longwave band. The above facts are indicative of a substantial redistribution of the electron density in their π-conjugated macrobicyclic frameworks as the result of a passing from the electron-withdrawing inherent chlorine atoms in molecule **9** to the electron-donating amine group(s) in the molecules of its obtained mono- and diaminemonoclathrochelate derivatives, and those of the further transformation of their propargylamine group(s) into the carborane-terminated *1H*-[1,2,3]-triazolylmethylamine ribbed substituents as well. The UV-ranges of the spectra of the obtained iron(II) clathrochelates contain a series of π–π*-transition bands characteristic of their polyazomethynemacrobicyclic frameworks; the bands of the same nature are also observed in the spectra of the iron(II) carboranoclathrochelates. The characteristic band with a maximum at approximately 270 nm of the carborane-terminated *1H*-[1,2,3]-triazolyl-containing fragment of the obtained hybrid molecules also appeared in the UV range of their spectra. Indeed, if an initial carborane is UV-silent in the spectral range under study (230–400 nm), the spectrum of 1-[(*o*-carboran-1’-yl)methyl]-4-pentyl-1,2,3-triazole, used as a model compound, contains in this range a low-intensive band with a maximum at 264 nm (see Appendix A); a more intensive band in its UV-spectrum appeared at 220 nm. The former characteristic band of a carborane-containing ribbed substituent is observed at approximately 270 nm in the spectra of the hybrid iron(II) carboranoclathrochelates under study.

The bulky molecular structures of the obtained polytopiciron(II) carboranoclathrochelates, combined with a flexibility of their ribbed functionalized substituents in a macrobicyclic cage framework, make their crystallization hardly possible. However, we succeeded in growing the single crystal **15** ∙ C_6_H_6_ from a solution of the corresponding hybrid complex in a benzene-*iso*-octane mixture. This crystal was suitable for collection of the X-ray diffraction data using synchrotron irradiation, but its quality did not allow us to perform a detailed discussion of the bond distances and the bond angles in the molecule **15**. However, its general view (Figure 1) and its crystal packing can be discussed. The encapsulated iron(II) ion is situated almost in the center of the Fe*N*_6_-coordination polyhedron possessing the geometry intermediate between a trigonal prism (TP, the distortion angle *φ =* 0°) and a trigonal antiprism (TAP, *φ =* 60°) with *φ* equal to 28. The diethylamine ribbed substituent and the terminal carborane of the ditopic molecule **15** are equiprobably disordered over two sites. The polyhedron conformation is strongly affected by the intramolecular dihydrogen bonding: a flexibility of the carborane-terminated ribbed substituent of this molecule allowed a formation of numerous C–H…H–B intramolecular interactions. The H(C) atom of the above carborane core also forms the intermolecular C–H…F–B interaction with an adjacent carboranoclathrochelate molecule (Figure 2). This allowed us to undoubtedly assign carbon and boron atoms of this core even without taking into account the corresponding bond distances. Another valuable intermolecular interaction in the X-rayed crystal **15** ∙ C_6_H_6_ is the N–H…N hydrogen bonding between the above diethylamine groups of one hybrid molecule and the heterocyclic five-membered 1H-[1,2,3]-triazolyl fragment of the second molecule of this type, thus giving the corresponding H-bonded carboranoclathrochelate dimers shown in Figure 2.

## 3. Experimental

### 3.1. General Information

The reagents used (FeCl_2_ · 4H_2_O, propargylamine, triethylamine, morpholine, diethylamine and copper acetate(II), sorbents and organic solvents were obtained commercially (SAF); *ortho*-carborane-1-methylazide, the dichloroclathrochelate precursor **9** and its dipropargylamine derivative **16** were prepared as described elsewhere [18,31,35,36].

Analytical data (C, H, N contents) were obtained with a CarloErba model 1106 microanalyzer.

MALDI-TOF mass spectra were recorded with and without the matrix using a MALDI-TOF-MS BrukerAutoflex II (BrukerDaltonics) mass spectrometer in reflecto-mol mode. The ionization was induced by a UV-laser with wavelength 337 nm. The samples were applied to a nickel plate; 2,5-dihydroxybenzoic acid was used as the matrix. The accuracy of measurements was 0.1%.

The ^1^H, ^1^H{^11^B}, ^11^B, ^11^B{^1^H}, ^19^F{^1^H} and ^13^C{^1^H}-NMR spectra were recorded from the solutions in CD_2_Cl_2_ with BrukerAvance 400 and Avance 600 spectrometers. The ^1^H and ^13^C{^1^H}-NMR measurements were performed using the residual signals of this deuterated solvent. ^11^B-NMR spectra were referred by the external BF_3_·O(C_2_H_5_)_2_; ^19^F-NMR spectra were referred by the external CFCl_3_.

IR spectra of the solid samples (KBr tablets) in the range 400–4000 cm^−1^ were recorded with a Perkin Elmer FT-IR Spectrum BX II spectrometer.

UV–vis spectra of the solutions in dichloromethane were recorded in the range 230–800 nm with a Varian Cary 50 spectrophotometer. The individual Gaussian components of these spectra were calculated using the Fityk program [37].

### 3.2. Synthesis

*FeBd_2_(HGmSpCarb)(BF)_2_* (**8**). This model spacer-containing carboranoclathrochelate was obtained using the two-step synthetic procedure that was first developed by [29].

#### 3.2.1. Preparation of the Clathrochelate Precursor FeBd_2_(HGmProp)(BF)_2_ (**7**)

Complex FeBd_2_(HGmCl)(BF)_2_ (0.49 g, 0.67 mmol) was dissolved/suspended in acetonitrile (5 mL), and a solution of a small excess of propargylamine (60 µL, 0.81 mmol) in acetonitrile (15 mL) was added dropwise to the stirring solution/suspension for 1.5 h under argon; the reaction course was controlled by TLC (SiO_2_ foil; eluent: dichloromethane –hexane 4:1 mixture). The reaction mixture was stirred overnight and then the dark-red solution evaporated to dryness. The solid residue was extracted with dichloromethane (25 mL), the extract was washed with water (40 mL, in two portions), dried with CaCl_2_ and filtered. The filtrate was evaporated to a half volume and then flash chromatographically separated on silica gel (30 mm layer, eluent: dichloromethane). The major dark-red elute was collected, evaporated to a small volume and precipitated with hexane. The precipitate was filtered off, washed with hexane (10 mL) and dried in vacuo. Yield: 0.36 g (0.492 mmol, 73%). M.w: 731.07 g/mol. Anal. Calc. for C_33_H_25_N_7_B_2_F_2_FeO_6_ (%): C, 54.17; H, 3.42; N, 13.41. Found (%): C, 54.06; H, 3.30; N, 13.42. ^1^H-NMR (CD_2_Cl_2_) δ 2.51 (s, 1H, HC≡C), 4.08 (s, 2H, CH_2_N), 5.96 (s, 1H, NH), 7.38 (m, 20H, Ph), 8.11 (s, 1H, HC=N). ^13^C{^1^H}-NMR (CD_2_Cl_2_) δ 33.25 (s, CH_2_N), 73.38 (s, HC≡C), 78.95 (s, HC≡C), 128.04, 129.36, 130.01, 130.57 (all s, Ph), 138.41 (s, HC=N), 149.74 (s, NHC=N), 155.68, 156.80 (both s, PhC=N). ^11^B{^1^H}-(CD_2_Cl_2_) −3.74 (d, *^1^J_B–F_* = 15.8 Hz). ^19^F{^1^H} (CD_2_Cl_2_) −168.42 (m). MS (MALDI-TOF) *m/z*: 732 [M + H^+^]^+^. IR (KBr), ν/cm^−1^: 1552 ν(C=N), 695, 1361 ν(HC≡C), 927m, 999, 1059, 1121 ν(N–O), 1204m ν(B–O) + ν(B–F). UV-Vis (CH_2_Cl_2_): λ_max_/nm (ε·× 10^−3^ mol^−1^·L·cm^−1^) 251(24), 278(14), 292(9.3), 335(3.3), 390(2.8), 470(24), 504(8.7).

#### 3.2.2. “Click” Reaction of the Clathrochelate Precursor FeBd_2_(HGmProp)(BF)_2_ (**7**)

Copper acetate(II) (0,0015 g, 0,0082 mmol) was dissolved/suspended in toluene (3 mL), and a solution of *ortho*-carborane-1-methylazide (0.06 g, 0.03 mmol) in toluene (0.8 mL) and the complex **7** (0.02 g, 0.027 mmol) were added to the stirring solution/suspension under argon. The reaction mixture was stirred at 70° for 2 h; the reaction course was controlled by TLC (SiO_2_ foil; eluent: methanol). Then the reaction mixture was cooled to room temperature and evaporated to dryness. The solid residue was separated by column chromatography on silica gel (eluent: methanol). The solid product was washed with hexane (15 mL), diethyl ether (15 mL) and dried in vacuo. Yield: 0.017 g (0.018·mmol, 67%). M.w.: 930.33 g/mol. Anal. Calc. for C_36_H_38_N_10_B_12_F_2_FeO_6_ (%): C, 46.48; H, 4.12; N, 15.06. Found (%): C, 46.66; H, 4.24; N, 15.22. ^1^H{^11^B}-NMR (CD_2_Cl_2_) δ 2.09 (br. s, 2H, carborane B–H), 2.21 (br. s, 6H, carborane B–H), 2.26 (br. s, 1H, B9), 2.42 (br. s, 1H, B12), 3.95 (s, 1H, carborane C–H), 4.71 (s, 2H, CH_2_), 5.06 (s, 2H, CH_2_), 6.15 (s, 1H, NH), 7.36 (m, 20H, Ph), 7.68 (s, 1H, 5H-triazole), 8.17 (s, 1H, N=C–H). ^11^B{^1^H}-NMR (CD_2_Cl_2_) δ −12.39 (s, 4B, B3, B6, B7, B11), −11.89 (s, 2B, B4, B5), −9.44 (s, 2B, B8, B10), −4.44 (s, 1B, B9), −1.34 (s, 1B, B12), −3.67 (m, 2B, B–F). ^19^F{^1^H}-NMR (CD_2_Cl_2_) δ −168.51 (m, 1F, B–F), −167.59 (m, 1F, B–F). MS (MALDI-TOF) *m/z* (*I*, %): 930 (100) [M]^+^, 953(40) [M + Na^+^]^+^, 969(10) [M + K^+^]^+^, 993 (35) [M + Cu^+^]^+^. IR (KBr), ν/cm^−1^: 931m, 1001, 1057, 1119 ν(N–O), 1196m ν(B–O) + ν(B–F), 1361 ν(N=N), 1580 ν(NC=N), 1635 ν(C=C), 2591 ν(B–H), 3392 ν(N–H. UV-Vis (CH_2_Cl_2_): λ_max_/nm (ε·× 10^−3^ mol^−1^·L·cm^−1^) 244(35), 265(1.0), 286(2.1), 295(3.5), 308(3.3), 336(0.5), 362(4.0), 479(24), 581(0.6).

*FeBd_2_(ClGmMorph)(BF)_2_* (**10**). For the first time, the preparation of this complex is described [38]. Complex **9** (0.12 g, 0.16 mmol) was dissolved in a dichloromethane–acetonitrile 1:1 mixture (6 mL) and morpholine (0.06 mL, 0.69 mmol) was added. The reaction mixture was stirred for 4 h and an additional portion of morpholine (0.06 mL, 0.69 mmol) was added; the reaction course was controlled by TLC (SiO_2_ foil, eluent: dichloromethane–hexane 9:1 mixture). The obtained solution was left at room temperature for 24 h, then the precipitate formed was filtered off, washed with water (20 mL, in two portions) and dried over CaCl_2_. The solid product was extracted with dichloromethane (4 mL) and the extract was separated by column chromatography on silica gel (eluent: dichloromethane–hexane 9:1 mixture). The first elute was discarded, and the second major elute was collected and evaporated to dryness. The solid residue was washed with hexane (15 mL), diethyl ether (15 mL) and dried in vacuo. Yield: 0.102 g (0.127·mmol, 85%). M.w.: 797.55 g/mol. Anal. Calc. for C_34_H_28_B_2_ClF_2_FeN_7_O_7_: C, 49.97; H, 3.73; N, 12.39. Found (%): C, 51.20; H, 3.54; N, 12.29. ^1^H-NMR (CD_2_Cl_2_*δ,* ppm): 3.62 (t, 4H, CH_2_N), 3.82 (t, 4H, CH_2_O), 7.36 (m, 20H, Ph). MS (MALDI-TOF): *m/z*: 797(64) [M]^+^^•^_,_ 820(29) [M + Na^+^]^+^, 836(7) [M + K^+^]^+^.^13^C{^1^H}-NMR (CD_2_Cl_2_*δ,* ppm): 49.8 (s, CH_2_N), 67.0 (s, CH_2_O), 127.9, 129.1, 130.1, 130.4 (two s, Ph), 131.3 (s, ClC=N), 149.3 (s, NC=N), 156.9, 157.8 (two s, PhC=N). UV-vis (CH_2_Cl_2_): λ_max_, nm (ε × 10^−3^, mol^−1^L cm^−1^): 240(24), 264(3.4), 279(18), 288(3.8), 342(3.1), 415(1.7), 477(17), 503(4.2).

*FeBd_2_(ClGmDea)(BF)_2_* (**11**). For the first time, the preparation of this complex is described [30]. Complex **9** (0.12 g, 0.16 mmol) was dissolved in dichloromethane (5 mL) and diethylamine (0.075 mL, 0.88 mmol) was added. The reaction mixture was stirred for 4 h; the reaction course was controlled by TLC (SiO_2_ foil, eluent: dichloromethane). The precipitate formed was filtered off, washed with water (20 mL, in two portions) and dried over CaCl_2_. The solid product was extracted with dichloromethane (4 mL) and the extract was separated by column chromatography on silica gel (eluent: dichloromethane–hexane 9:1 mixture). The first elute was discarded, and the second major elute was collected and evaporated to dryness. The solid residue was washed with hexane (15 mL), diethyl ether (15 mL) and dried in vacuo. Yield: (0.126 mmol, 79%). M.w.: 783.57 g/mol. Anal. Calc. for C_34_H_30_B_2_ClF_2_FeN_7_O_6_: C, 52.51; H, 3.82; N, 12.48. MS (MALDI-TOF): *m/z*: 783 [M]^+^^•^. ^1^H-NMR (CD_2_Cl_2_*δ,* ppm): 1.24 (t, 6H, NCH_2_CH_3_), 3.53 (q, 4H, NCH_2_CH_3_), 7.38 (m, 20H, Ph). ^13^C{^1^H}-NMR (CD_2_Cl_2_*δ,* ppm): 44.79 (s, NCH_2_CH_3_), 45.78 (s, NCH_2_CH_3_), 127.9, 129.2, 129.3, 130.5 (two s, Ph), 130.9 (s, ClC=N), 151.1 (s, NC=N), 156.8, 157.7 (two s, PhC=N). UV-vis (CH_2_Cl_2_): λ_max_, nm (ε × 10^−3^, mol^−1^L cm^−1^): 242(31), 265(4.0), 287(1.5), 288(16), 337(3.8), 416(2.1), 481(24), 543(1.7).

*FeBd_2_(PropGmMorph)(BF)_2_* (**12**). Complex **10** (0.16 g, 2 mmol) was dissolved in DMF (4 mL) and propargylamine (0. 12 mL, 1.8 mmol) was added. The reaction mixture was heated at 110 °C for 4 h and then left for 24 h; the reaction course was controlled by TLC (SiO_2_ foil, eluent: methanol–dichloromethane 9:1 mixture). The obtained solution was precipitated with 1.4M NaClO_4_ aqueous solution, the precipitate formed was filtered off, washed with water (50 mL, in five portions), methanol (5 mL), diethyl ether (5 mL) and hexane (30 mL, in 3 portions), and dried in air. The solid product was extracted with dichloromethane (4 mL) and the extract was separated by column chromatography on silica gel (eluent: methanol-dichloromethane 9:1 mixture). The first major elute was collected and evaporated to dryness. The solid residue was washed with hexane (15 mL), diethyl ether (15 mL) and dried in vacuo. Yield: 0.11 g (0.134 mmol, 73%). M.w.: 816.17 g/mol. Anal. Calc. for C_37_H_32_B_2_F_2_FeN_8_O_7_: C, 54.35; H, 4.10; N, 13.84. MS (MALDI-TOF): *m/z*: 816 [M]^+^^• 1^H-NMR (CD_2_Cl_2_, *δ,* ppm): 2.46 (s, 1H, NHCH_2_CCH), 3.56 (t, 4H, CH_2_N), 3.83 (t, 4H, CH_2_O), 4.41 (d, 2H, NHCH_2_CCH), 5.84 (t, 1H, NHCH_2_CCH), 7.36 (m, 20H, Ph).^13^C{^1^H}-NMR (CD_2_Cl_2_*δ,* ppm): 34.1 (s, NHCH_2_CCH), 49.2 (s, CH_2_N), 67.0 (s, CH_2_O), 72.8 (s, NHCH_2_CCH), 79.7 (s, NHCH_2_CCH), 126.1, 127.8, 129.7, 130.5 (two s, Ph), 149.6, 157.0 (two s, NC=NH), 157.1 (two s, PhC=N). UV-vis (CH_2_Cl_2_): λ_max_,nm(ε × 10^−3^, mol^−1^L cm^−1^): 237(24), 251(0.6), 278(20), 298(3.8), 361(3.3), 387(22), 429(6.9), 499(21), 528(1.0), 563(1.7).

*FeBd_2_(PropGmDea)(BF)_2_* (**13**). Complex **11** (0.16 g, 0.2 mmol) was dissolved in DMF (4 mL) and propargylamine (0. 12 mL, 1.8 mmol) was added. The reaction mixture was heated at 110 °C for 4 h and then left for 24 h at room temperature; the reaction course was controlled by TLC (SiO_2_ foil, eluent: dichloromethane-hexane 9:1 mixture). The obtained solution was precipitated with 5% aqueous hydrochloric acid (2.5 mL), the precipitate formed was filtered off, washed with water (50 mL, in five portions), methanol (20 mL, in two portions), diethyl ether (30 mL, in three portions) and hexane (30 mL, in three portions). The solid product was extracted with dichloromethane (4 mL) and the extract was separated by column chromatography on silica gel (eluent: dichloromethane–hexane 20:1 mixture). The first elute was discarded, and the second major elute was collected and evaporated to dryness. The solid residue was washed with hexane (15 mL), diethyl ether (15 mL) and dried in vacuo. Yield: 0.113 g (0.141 mmol, 69%). M.w.: 802.19 g/mol. Anal. Calc. for C_37_H_34_B_2_F_2_FeN_8_O_6_: C, 55.40; H, 4.27; N, 13.97. Found (%): C, 55.20; H, 4.35; N, 13.87. MS (MALDI-TOF): *m/z*: 802 [M]^+^^•^. ^1^H-NMR (CD_2_Cl_2_*δ,* ppm):1.17 (t, 6H, NCH_2_CH_3_), 2.44 (s, 1H, NHCH_2_CCH) 3.44 (q, 4H, NCH_2_CH_3_), 4.47 (d, 2H, NHCH_2_CCH), 5.87 (t, 1H, NHCH_2_CCH), 7.34 (m, 20H, Ph). ^13^C{^1^H}-NMR (CD_2_Cl_2_*δ,* ppm): 13.1 (s, NCH_2_CH_3_), 34.2 (s, NHCH_2_CCH), 45.0 (s, NCH_2_CH_3_), 72.5 (s, NHCH_2_CCH), 79.8 (s, NHCH_2_CCH), 127.9, 129.4, 129.6, 130.5 (all two s, Ph), 149.6, 157.2 (two s, NC=NH), 157.5 (s, PhC=N). UV-vis (CH_2_Cl_2_): λ_max_, nm (ε × 10^−3^, mol^−1^L cm^−1^): 243(31), 295(18), 338(2.1), 379(5.0), 434(5.1), 498(19), 525(1.8), 557(1.9).

*FeBd_2_(MorphGmSpCarb)(BF)_2_* (**14**). Copper acetate(II) (0.005 g, 0.027 mmol) was dissolved/suspended in toluene (3 mL) and a solution of *ortho*-carborane-1-methylazide (0.02 g, 0.1 mmol) in toluene (1 mL) and the complex **12** (0.041 g, 0.05 mmol) were added under argon to the stirring solution/suspension. The reaction mixture was stirred at 80 °C for 4 h, then cooled to room temperature and evaporated to dryness. The solid residue was separated by column chromatography on silica gel (eluent:dichloromethane–methanol 20:1 mixture). The first major elute was collected and evaporated to dryness. The solid residue was washed with hexane (15 mL), diethyl ether (15 mL) and dried in vacuo. Yield: 0.032 g (0.031 mmol, 68%). M.w.: 1015.43 g/mol. Anal. Calc. for C_40_H_45_B_12_F_2_FeN_11_O_7_: C, 47.31; H, 4.47; N, 15.17. Found (%): C, 47.33; H, 4.46; N, 15.48. MS (MALDI-TOF): *m/z*: 1016(45) [M]^+^^•^_,_ 1039(30) [M + Na^+^]^+^, 1056(38) [M + K^+^]^+^. ^1^H-NMR (CD_2_Cl_2_, *δ,* ppm): 2.06 (br. s, 2H, carborane B–H), 2.20 (br. s, 6H, carborane B–H), 2.27 (br. s, 1H, B9), 2.40 (br. s, 1H, B12), 3.56 (t, 4H, CH_2_N), 3.83 (t, 4H, CH_2_O), 5.03 (d, 2H, NHCH_2_CC), 5.60 (t, 1H, NHCH_2_CC), 7.40 (m, 20H, Ph), 7.78 (s, 1H, 5H-triazole). ^13^C{^1^H}-NMR (CD_2_Cl_2_*δ,* ppm): 13.92 (s, 2C, carborane), 29.7 (s, NHCH_2_CC), 40.06 (s, NHCH_2_CC), 49.3 (s, CH_2_N), 67.2 (s, CH_2_O), 71.5 (s, NHCH_2_CC), 127.9, 129.5, 129.8, 130.5 (two s, Ph), 146.1, 147.8 (two s, NC=NH), 156.9 (two s, PhC=N).^11^B{^1^H}-NMR (CD_2_Cl_2_) δ −12.17 (s, 4B, B3, B6, B7, B11), −11.21 (s, 2B, B4, B5), −9.96 (s, 2B, B8, B10), −4.83 (s, 1B, B9), −2.06 (s, 1B, B12), −3.65 (m, 2B, B–F). ^19^F{^1^H}-NMR (CD_2_Cl_2_) δ −167.96 (m, 1F, B–F), −166.41 (m, 1F, B–F). UV-vis (CH_2_Cl_2_): λ_max_, nm (ε × 10^−3^, mol^−1^L cm^−1^): 243(28), 270(0.5), 295(11), 313(4.1), 382(0.9), 433(1.3), 474(7.5), 501(12), 525(1.3).

*FeBd_2_(DeaGmSpCarb)(BF)_2_* (**15**). Copper acetate(II) (0.005 g, 0.027 mmol) was dissolved/suspended in toluene (3 mL) and a solution of *ortho*-carborane-1-methylazide (0.015 g, 0.075 mmol) in toluene (0.8 mL) and the complex **13** (0.02 g, 0.024 mmol) were added under argon to the stirring solution/suspension. The reaction mixture was stirred at 80 °C for 2 h; the reaction course was controlled by TLC (SiO_2_ foil; eluent: dichloromethane). Then the reaction mixture was cooled and evaporated to dryness. The solid residue was separated by column chromatography on silica gel (eluent: dichloromethane–methanol 20:1 mixture). The first major elute was evaporated to dryness, washed with hexane (15 mL), diethyl ether (15 mL) and dried in vacuo. Yield: 0.019 g (0.019 mmol, 65%). M.w.: 1001.45 g/mol. Anal. Calc. for C_40_H_47_B_12_F_2_FeN_11_O_6_: C, 47.97; H, 4.73; N, 15.39. Found (%): C, 47.72; H, 4.86; N, 15.22. MS (MALDI-TOF): *m/z*: 1002(45) [M]^+^^•^_,_ 1025(100) [M + Na^+^]^+^, 1040(55) [M + K^+^]^+^. ^1^H{^11^B}-NMR (CD_2_Cl_2_) δ: 1.16 (s, 6H, NCH_2_CH_3_), 2.01 (br. s, 2H, carborane B–H), 2.16 (br. s, 6H, carborane B–H), 2.35 (br. s, 1H, B9), 2.59 (br. s, 1H, B12), 3.41 (m, 4H, NCH_2_CH_3_) 3.92 (s, 1H, carborane C–H), 4.75 (s, 2H, CH_2_), 5.00 (s, 2H, CH_2_), 6.18 (s, 1H, NH), 7.38 (m, 20H, Ph), 7.79 (s, 1H, 5H-triazole). ^11^B{^1^H}-NMR (CD_2_Cl_2_) δ −12.40 (s, 4B, B3, B6, B7, B11), −11.31 (s, 2B, B4, B5), −9.86 (s, 2B, B8, B10), −4.91 (s, 1B, B9), −2.03 (s, 1B, B12), −3.61 (m, 2B, B–F). ^19^F{^1^H}-NMR (CD_2_Cl_2_) δ −168.14 (m, 1F, B–F), –166.42 (m, 1F, B–F). UV-vis (CH_2_Cl_2_): λ_max_,nm(ε × 10^−3^, mol^−1^L cm^−1^): 241(29), 270(0.8), 292(14), 338(3.9), 385(0.5), 435(2.4), 489(7.0), 498(10), 524(2.9).

*FeBd_2_(Gm(SpCarb)_2_)(BF)_2_* (**17**). Copper acetate(II) (0,0028 g, 0,015 mmol) was dissolved/suspended in toluene (3 mL), and a solution of *ortho*-carborane-1-methylazide (0.01 g, 0.05 mmol) in toluene (0.8 mL) and the complex **16** (0.02 g, 0.0255 mmol) were added under argon to the stirring solution/suspension. The reaction mixture was stirred at 70° for 2 h; the reaction course was controlled by TLC (SiO_2_ foil; eluent: methanol). Then the reaction mixture was cooled to room temperature and evaporated to dryness. The solid residue was separated by column chromatography on silica gel (eluent: methanol). The major elute was evaporated to dryness, the solid residue was washed with hexane (15 mL), diethyl ether (15 mL) and dried in vacuo. Yield: 0.012 g (0.010 mmol, 61%). M.w.: 1182.65 g/mol. Anal. Calc. for C_42_H_54_B_22_F_2_FeN_14_O_6_: C, 42.66; H, 4.60; N, 16.58. Found (%): C, 45.52; H, 4.43; N, 16.39. MS (MALDI-TOF): *m/z*: 1184(14) [M]^+•^_,_ 1206(5) [M + Na^+^]^+^, 1246(81) [M + K^+^ + Na^+^]^+ 1^H{^11^B}-NMR (CD_2_Cl_2_) δ 2.00 (br. s, 4H, carborane B–H), 2.35 (br. s, 12H, carborane B–H), 2.38 (br. s, 2H, B9), 2.46 (br. s, 2H, B12), 3.68 (s, 2H, carborane C–H), 4.75 (s, 4H, CH_2_), 5.08 (s, 4H, CH_2_), 6.23 (s, 2H, NH), 7.40 (m, 20H, Ph), 7.73 (s, 2H, 5H-triazole). ^11^B{^1^H}-NMR (CD_2_Cl_2_) δ −12.39 (s, 8B, B3,B3′, B6,B6′, B7,B7′, B11,B11′), −11.78 (s, 4B, B4,B4′, B5,B5′), −9.21 (s, 4B, B8,B8′, B10,B10′), −4.44 (s, 2B, B9,B9′), −1.38 (s, 2B, B12,B12′), −3.67 (m, 2B, B–F). UV-vis (CH_2_Cl_2_): λ_max_,nm(ε × 10^−3^, mol^−1^L cm^−1^): 240(24), 281(16), 365(6.5), 433(1.6), 466(1.8), 515(13), 616(1.6).

### 3.3. X-ray Crystallography

The single crystals of the complex **15** ∙ C_6_H_6_ were grown from a solution of the corresponding carboranoclathrochelate in a benzene–*iso*-octane 1:2 mixture. Intensities of the reflections were collected at the K4.4 “Belok” beamline of the Kurchatov Synchrotron Radiation Source (NRC “Kurchatov Institute”, Moscow, Russia) at a wavelength of 0.79313 Å using a Rayonix CCD 165 detector, but even its best available single crystal was characterized by large R_int_ and converged to poor R_1_. Nevertheless, the quality of the collected data is clearly sufficient to evaluate a connectivity of the molecular structure of **15**. Data collection was performed at a low temperature of 100 K using an Oxford CryoJet (Oxford Cryosystems) at λ = 0.79313 Å using *φ*-scans. At 100 K, the crystal C_46_H_53_B_12_F_2_FeN_11_O_6_ (Fw = 1079.56) was monoclinic, space group *C2/c*, *a* = 39.070(8), *b* = 12.070(2), *c* = 23.340(5) Å, β = 105.99(3)°, *V* = 10581(4) Å^3^, *d_calc_* = 1.355 g cm^–3^, μ = 0.468 mm^–1^, *F(000)* = 4464. The image integration was performed using the iMosflm software [39]. The integrated intensities were empirically corrected for an absorption using the Scala program [40]. The structure was solved by the SHELXT method [41] and refined by full-matrix least squares against *F^2^.* Non-hydrogen atoms were refined anisotropically except those of the disordered fragments. The carborane polyhedron and the diethylamine substituent of the molecule **15**, as well as the benzene solvate molecule, are equiprobably disordered over two sites; therefore, their carbon and boron atoms were refined isotropically. One DFIX (the B6A–B5 distance was fixed as 1.8 Å) and several SADI instructions (all the B–C distances are equal) were applied to refine the disordered carborane core. Although these atoms could not be distinguished based on either the obtained Rint and thermal ellipsoid values or the bond distances, the close C–H...H–B and C–H...F intramolecular and intermolecular interactions between atoms with various atomic charges are more likely to form than the C–H...H–C and B–H...F interactions between atoms with similar atomic charges. This fact allowed us to distinguish carbon and boron atoms in this carborane polyhedron. Positions of hydrogen atoms were calculated and all hydrogen atoms were included in a refinement by the riding model with *U_iso_*(H) = 1.2*U_eq_*(X). All calculations were made using the SHELXL2014 [42] and OLEX2 [43] program packages. Refinement was converged to R_1_ = 0.1269 and GOF = 1.073 (4235 observed reflections), wR_2_ = 0.2890 (9157 independent reflections, R^int^ = 0.1231). CCDC 2002890 contains the supplementary crystallographic data for this paper. These data can be obtained free of charge via http://www.ccdc.cam.ac.uk/structures/ (accessed on 9 June 2021).

## 4. Conclusions

Using a developed synthetic approach based on a combination of the 1,3-dipolar cycloaddition “click” reaction with the stepwise nucleophilic substitution of an appropriate halogenoclathrochelate precursor, we succeeded in the synthesis of spacer-containing di- and tritopic iron(II) carboranoclatrochelates, formed by a covalent linking between their different polyhedral(cage) fragments of the same polytopic molecule. Their chemical design allows regulation of the hydrophilic–hydrophobic balance in their hybrid molecules using various clathrochelate (as a more polar core)-to-carborane (as a more hydrophobic core) stoichiometries. The presence of a biorelevant substituent in these molecules may allow their efficient supramolecular binding to various biological targets. Further searches among a series of the polytopic and multicage covalent conjugates—the structurally flexible iron(II) carboranoclatrochelates—will be used for evaluation of the hybrid complexes possessing an appropriate molecular structure, the optimal physical characteristics and the physicochemical properties, for their testing as prospective prodrugs for drug therapy of various diseases and/or boron-enriched compounds for ^10^B-NCT treatment.

## Data Availability

CCDC 2002890 contains the supplementary crystallographic data for this paper. These data can be obtained free of charge via http://www.ccdc.cam.ac.uk/structures/ (accessed on 9 June 2021).

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
