# Peer review of "Synthesis and Structure of the Bis- and Tris-Polyhedral Hybrid Carboranoclathrochelates with Functionalizing Biorelevant Substituents—The Derivatives of Propargylamine Iron(II) Clathrochelates with Terminal Triple C≡C Bond(s)"

_molecules, 2021, doi:10.3390/molecules26123635_

Round 1

Reviewer 1 Report

The manuscript submitted by Voloshin et al. describes the preparation of new hybrid carboranoclathrochlates via post-synthetic modifications of their macrobicyclic precursors. The manuscript is well written, the main idea of it is attractive, and the obtained results and their interpretation & discussion deserve publication in Molecules after minor revision.

Title:

Page 1, line 3 as well as “Citations”: should be «biorelevant» instead of “biorelevaunt”

Introduction part:

Page 2, line 54: A curly bracket should be after “carboxyl)”.

Results and Discussion part:

Page 7, line 235: please check the range of the MLCT bands for the compounds 10 and 11; “in the range 500 – 500 nm” seems to be a mistake;

Page 12, lines 487: The authors suggest that the carborane polyhedron is disordered and the quality of XRD experiment is not very high. So, how its boron and carbon atoms have been assigned and refined?

Experimental part:

Page 9, lines 235 and 332: UV-Vis data for the complex 7 are presented two times.

Page 10, line 350: IR spectrum is given for the compound 7, but in the Section 3.1. there is no description of the corresponding instrument.

Page 11, lines 400 and 436: Triplet signal in NMR spectra is marked in Russian.

Page 11, lines 433 and 453: it is better to give the yields in mg.

List of the References:

43% of the authors’s self-citations is found in this List. I guess, this is too much percentage.

Author Response

We would like to thank you for helpful suggestions on how to improve our manuscript our article entitled “Synthesis and structure of the bis- and tris-polyhedral hybrid carboranoclathrochelates with functionalizing biorelevant substituents – the derivatives of propargylamine iron(II) clathrochelates with terminal triple CºC bond(s)” by G.E. Zelinskii, I.P. Limarev, A.V. Vologzhanina, V.A. Olshevskaya, A.V. Makarenkov, P.V. Dorovatovskii, A.S. Chuprin, M.A. Vershinin, S.V. Dudkin, Y.Z. Voloshin. We made all the corrections suggested by Reviewer, as is explained below in more details.

Comment of Reviewer 1:

“Title: Page 1, line 3 as well as “Citations”: should be «biorelevant» instead of “biorelevaunt””

Authors’ reply:

We have corrected the title as instructed by Reviewer 1.

“Introduction part: Page 2, line 54: A curly bracket should be after “carboxyl)”.”

Authors’ reply:

We have corrected the text as instructed by Reviewer 1.

“Results and Discussion part: Page 7, line 235: please check the range of the MLCT bands for the compounds 10 and 11; “in the range 500 – 500 nm” seems to be a mistake;”

Authors’ reply:

We have corrected the text as instructed by Reviewer 1.

“Page 12, lines 487: The authors suggest that the carborane polyhedron is disordered and the quality of XRD experiment is not very high. So, how its boron and carbon atoms have been assigned and refined?”

Authors’ reply:

Although these atoms could not be distinguished basing on the obtained Rint and thermal ellipsoids values or the bond distances as well, the close C–H...H–B and C–H...F intramolecular and intermolecular interactions between atoms with various atomic charges are more likely to form than the C–H...H–C and B–H...F interactions between atoms with similar atomic charges. This fact allowed us to distinguish carbon and boron atoms in this carborane polyhedron.

“Experimental part: Page 9, lines 235 and 332: UV-Vis data for the complex 7 are presented two times.”

Authors’ reply:

We have corrected the text as instructed by Reviewer 1.

Page 10, line 350: IR spectrum is given for the compound 7, but in the Section 3.1. there is no description of the corresponding instrument.”

Authors’ reply:

We have added the description of the used IR spectrometer.

“Page 11, lines 400 and 436: Triplet signal in NMR spectra is marked in Russian.”

Authors’ reply:

We have corrected the text as instructed by Reviewer 1.

“Page 11, lines 433 and 453: it is better to give the yields in mg.”

Authors’ reply:

In our opinion, because the yields of all the target complexes, as well as those of the initial reagents are given in grams, we suggest that reagents the yields of these two compounds should also be given in the same units.

“List of the References: 43% of the authors’s self-citations is found in this List. I guess, this is too much percentage.”

Authors’ reply:

We have corrected the List of References trying to reduce the percentage of self-citation. In the present version of ms, its level is approximately 34%.

Reviewer 2 Report

In this paper the author describes the preparation and characterization of a set of compounds, two of them bearing and o-carborane cluster, which can be potential candidates for BNCT. The paper is well written; the introduction information is well focused on the topic of the paper. The experimental part is described in detail including the experimental synthesis of all the compounds and their complete characterization. The results and discussion are also well described. I have not found mistakes along the manuscript. The paper is worthy to be published in Moleculues after minor revision

-As a general point, I find that the authors have included a large number of self-citations, 15 references over a total of 33. On the other hand, I miss some recent references on carborane clusters that should be included:

1) Cells 2020, 9, 1408.

2) Cancer Commun. 2018, 38, 36.

3) Coord. Chem. Rev. 2020, 405, 213139.

4) Xuan, S.; Vicente, M. G. H. Recent Advances in Boron Delivery

Agents for Boron Neutron Capture Therapy (BNCT). In Boron-based

Compounds: Potential and Emerging Applications in Biomedicine; Hey-

Hawkins, E., Vinas, C., Eds.; John Wiley & Sons: Hoboken, NJ, 2018;

pp 298−342.

Author Response

We would like to thank you for helpful suggestions on how to improve our manuscript our article entitled “Synthesis and structure of the bis- and tris-polyhedral hybrid carboranoclathrochelates with functionalizing biorelevant substituents – the derivatives of propargylamine iron(II) clathrochelates with terminal triple CºC bond(s)” by G.E. Zelinskii, I.P. Limarev, A.V. Vologzhanina, V.A. Olshevskaya, A.V. Makarenkov, P.V. Dorovatovskii, A.S. Chuprin, M.A. Vershinin, S.V. Dudkin, Y.Z. Voloshin. We made all the corrections suggested by Reviewer, as is explained below in more details.

Comment of Reviewer 2:

“As a general point, I find that the authors have included a large number of self-citations, 15 references over a total of 33. On the other hand, I miss some recent references on carborane clusters that should be included:

1) Cells 2020, 9, 1408.

2) Cancer Commun. 2018, 38, 36.

3) Coord. Chem. Rev. 2020, 405, 213139.

4) Xuan, S.; Vicente, M. G. H. Recent Advances in Boron Delivery Agents for Boron Neutron Capture Therapy (BNCT). In Boron-based Compounds: Potential and Emerging Applications in Biomedicine; Hey-Hawkins, E., Vinas, C., Eds.; John Wiley & Sons: Hoboken, NJ, 2018; pp 298−342.”

Authors’ reply:

We have corrected the List of References trying to reduce the percentage of self-citation. In the present version of ms its level is approximately 34%; the above references have been also added to this List.

Reviewer 3 Report

The article entitled " Synthesis and structure of the bis- and tris-polyhedral hybrid carboranoclathrochelates with functionalizing biorelevaut substituents – the derivatives of propargylamine iron(II) clathrochelates with terminal triple CC bond(s)" described the synthesis of new carboraneclathrochelates via click chemistry.

They demonstrated first how the precursors were done, and were characterized. then they achieve the click reaction, showing by various analytical technics that they obtain the product. 

The article is well written and the compounds clearly characterized.

I think this article is suitable for Molecules and can be published.

Author Response

We would like to thank you for helpful suggestions on how to improve our manuscript.